# Promoting Topic Coherence and Inter-Document Consorts in Multi-Document Summarization via Simplicial Complex and Sheaf Graph

**Yash Kumar Atri[1], Arun Iyer[2], Tanmoy Chakraborty[3], Vikram Goyal[1]**
[1]IIIT-Delhi, India; [2]Microsoft Research, India; [3]IIT Delhi, India
{yashk,vikram}@iiitd.ac.in; ariy@microsoft.com; tanchak@iitd.ac.in

## Abstract

Multi-document Summarization (MDS) characterizes compressing information from multiple source documents to its succinct summary. An ideal summary should encompass all topics and accurately model cross-document relations expounded upon in the source documents. However, existing systems either impose constraints on the length of tokens during the encoding or falter in capturing the intricate cross-document relationships. These limitations impel the systems to produce summaries that are non-factual and unfaithful, thereby imparting an unfair comprehension of the topic to the readers. To counter these limitations and promote the information equivalence between the source document and generated summary, we propose FABRIC, a novel encoder-decoder model that uses pre-trained BART to comprehensively analyze linguistic nuances, simplicial complex layer to apprehend inherent properties that transcend pairwise associations and sheaf graph attention to effectively capture the heterophilic properties. We benchmark FABRIC with eleven baselines over four widely-used MDS datasets – Multinews, CQASumm, DUC and Opinosis, and show that FABRIC achieves consistent performance improvement across all the evaluation metrics (syntactical, semantical and faithfulness). We corroborate these improvements further through qualitative human evaluation. The source code is available at https://github.com/LCS2-IIITD/FABRIC

## 1 Introduction

Multi-document summarization (MDS) aims to formulate a summary that captures the essence and main points of multiple documents on a specific topic. In contrast to single document summarization (SDS) that focuses on generating summaries from a single source, MDS faces additional challenges such as dealing with a larger search space, redundant documents, and conflicting opinions. These challenges pose difficulties for deep learning models, often resulting in the generation of summaries that are the results of hallucination and they often lack faithfulness (Maynez et al., 2020). The development of Large Language Models (LLMs) such as BERT (Devlin et al., 2019), BART (Lewis et al., 2020), and T5 (Raffel et al., 2020) has significantly advanced the field of text summarization. However, generating factually accurate and faithful summaries remains a persistent challenge (OpenAI, 2023).

Recent studies have focused on enhancing the faithfulness and factuality of summarization models, which is broadly categorized into three types – (i) post editing models for correcting generated summaries (Fabbri et al., 2022; Balachandran et al., 2022), (ii) using multi-task problems like question-answering (Durmus et al., 2020; Deutsch et al., 2020), entailment (Roit et al., 2023; Pasunuru and Bansal, 2018) etc., and (iii) using external knowledge (Huang et al., 2020; Mao et al., 2022; Chen and Yang, 2021; Mao et al., 2022; Lyu et al., 2022) to support the model during summary generation. In contrast to these methods, we propose a novel approach that promotes topic coherence and inter-document connections, all without the need for additional parameters, post-editing, external knowledge, or auxiliary tasks.

Additionally, capturing semantic connections and multi-level representation can further aid the model in discerning the redundant and pivotal information in multiple documents. Graph neural networks (Scarselli et al., 2009) are constrained in terms of subpar performance in heterophilic settings[1] and oversmoothing (Song et al., 2023) when utilized with multi-layer neural networks. To mitigate these concerns, we propose FABRIC[2] in which we introduce simplicial complex (Giusti

---

[1]In a heterophilic setting, diverse node and edge types can complicate the message-passing process of GNN, therefore affecting its performance.

[2]FABRIC: FAirness using BaRt, sImplicial Complex and sheaf

et al., 2022) layer and sheaf graph (Hansen and Gebhart, 2020) attention for multi-document summarization. Simplicial complexes are employed to apprehend the interconnections among diverse elements of the text, encompassing words, phrases, or sentences. By representing these associations as simplices (geometric shapes formed by combining vertices, edges, and higher-dimensional counterparts), a simplicial complex furnishes a structure to examine the connectivity and coherence within the text. On the other hand, sheaf graphs facilitate the assimilation of diverse node types and attributes within the graph structure. Each node can symbolize a specific element, such as a word or a phrase, and convey its own attributes or features. By considering these heterogeneous characteristics, sheaf graphs can apprehend and model the relationships among distinct types of nodes. This empowers FABRIC to comprehend various elements and their relationships, enabling the generation of faithful and accurate summaries.

Moreover, when dealing with MDS, a critical challenge that neural networks often encounter is processing large documents. Many recent studies have attempted to concatenate multiple documents into a flat sequence and train SDS models on them (Fabbri et al., 2019). However, this approach fails to consider the inter-document relationship and the presence of redundant long input vectors. Even very large language models like Long-T5 (Guo et al., 2021) and BART-4096 (Lewis et al., 2020), which are capable of handling input vectors beyond 2000 tokens, face practical limitations due to the quadratic growth of input memory space. Furthermore, such extensive global attention may exhibit lower performance compared to alternative methods. To address this issue, we propose a novel approach, called *topic-assisted document segmentation*. It involves using a segment of the document as a context vector, compressing it into an oracle summary while covering major topics, and appending it to the next segment to generate the subsequent context vector. By employing this method, summarization models can learn the representation of the document from a compressed form, overcoming the challenges associated with processing large documents.

In short, our contributions are as follows.

1. We propose a novel topic assisted document segmentation method, which allows any language model to generate contextual vectors for any input length.

2. We propose FABRIC, a novel encoder-decoder model that uses pre-trained BART to comprehensively analyze linguistic nuances, Simplicial Complex layer to apprehend inherent properties that transcend pairwise associations and sheaf graph attention to more effectively apprehend the heterophilic properties.

3. We evaluate FABRIC using four standard metrics – ROUGE (Lin, 2004), BARTScore (Yuan et al., 2021), FactCC (Kryscinski et al., 2020), and SummaC (Laban et al., 2022), to assess the quantitative, and qualitative performances and faithfulness. We showcase that FABRIC fares well against eleven widely-popular abstractive baselines. FABRIC beats the best baseline, PRIMERA (Xiao et al., 2022) by +0.34 ROUGE-L on Multinews, +4.90 Rouge-L on CQASumm, +2.94 on DUC and +2.28 Rouge-L on the Opinosis dataset. Our qualitative analyses on semantic similarity by BARTScore and faithfulness by FactCC and SummaC also show considerable gains compared to the baseline-generated summaries. We further perform an exhaustive human evaluations and comparison with ChatGPT[3] to compare the quality of the system-generated summaries.

## 2 Related Works

**Abstractive summarization.** The task of text summarization has seen remarkable improvement with the introduction of attention (Bahdanau et al., 2015) and transformer (Vaswani et al., 2017) based approaches. Later, transformers coupled with content selection (Gehrmann et al., 2018; Atri et al., 2021), attention approximations (Zaheer et al., 2020), windowed and task-oriented attention (Beltagy et al., 2020) were used to improve the performance further. The introduction of LLMs has also shown incredible performance by fine-tuning them over few epochs. LMs such as BERT (Devlin et al., 2019), BART (Lewis et al., 2020), T5 (Raffel et al., 2020), and Pegasus (Zhang et al., 2020) are pre-trained over a vast corpus and later finetuned over the summarization datasets to achieve state-of-the-art results. Reinforcement learning (RL) algorithms have also been leveraged to further improve the performance of the models. In abstractive summarization, Lee and Lee (2017); Pasunuru and Bansal (2018); Böhm et al. (2019); Atri et al. (2023b) intro-

---

[3]https://openai.com/blog/chatgpt/

duced the notion of how RL-based models can aid in text summarization. Various reward measures like Rouge-L (Paulus et al., 2017), human feedback (Böhm et al., 2019) and combination of rewards (Pasunuru and Bansal, 2018) have been used.

**Faithfulness.** Existing methods for improving faithfulness include proposing new fact-corrected datasets (Balachandran et al., 2022), ensembling language models (Guo et al., 2022), adversarial techniques (Wu et al., 2022), target correction (Fabbri et al., 2022; Adams et al., 2022; Lee et al., 2022), Natural Language Inference (NLI) models (Laban et al., 2022), rejecting noisy tokens (Cao et al., 2022), controlling irrelevant sentences (Ghoshal et al., 2022), and contrasting candidate generation (Chen et al., 2021). Others use auxiliary information from source documents like additional input (Dou et al., 2021), entity (Zhang et al., 2022), knowledge graphs (Lyu et al., 2022; Huang et al., 2020; Zhao et al., 2021) or RL-based approaches with rewards as entailment feedback (Roit et al., 2023), and structured fact extraction (Zhang et al., 2021). Nevertheless, these methodologies either raise question upon the veracity of the datasets, cull out discordant data points or employ extrinsic data sources to alleviate the factual incongruities. Formulating a new dataset entails substantial costs, and winnowing existing ones might diminish the dataset pool for training a competitive model. In contrast, our proposed approach surmounts these constraints, capitalizing on the entirety of accessible data during training and leveraging the dynamics of simplicial complexes and sheaf graphs to apprehend the inter- and intra-document relationships, thereby engendering exceedingly faithful and factual summaries.

# 3 Proposed Methodology

We present FABRIC, a novel multi-encoder-decoder model for multi-document abstractive text summarization. Our approach harnesses the power of topic assisted document segments and leverages higher order topological spaces to adeptly extract essential information. Figure 1 shows a schematic diagram of FABRIC. This section explains each component of FABRIC in detail.

## 3.1 Topic Assisted Document Segments

Transforming the entire document to a contextual vector is very expensive pertaining to the quadratic growth of memory and computations in

Transformer-based models. LMs like Longformer (Beltagy et al., 2020) are able to process documents in a linear complexity. However, they fail to efficiently capture the context of the whole document. To address this limitation, we propose a new data modelling strategy, called *Topic Assisted Document Segmentation with Oracle proxies*. At first, the source document $D$ is split into segments $D_1, D_2, D_3, \ldots, D_n$ based on major topics identified. We use BERTopic (Grootendorst, 2022) for topical identification. Next, we formulate oracle summaries for segment $D_1$ using an Integer Linear Programming (ILP) solver (Gillick et al., 2008) and penalize it with BERTopic (Grootendorst, 2022) to aid the module in generating oracle summaries and to cover all major topics of the document. We append the oracle summary obtained $O_1$ to the next segment $O_1 + D_1$ and formulate the next oracle summary using only $O_1 + D_1$. Each generated segment is limited to 768 tokens for BART to process the input vector effectively. Each segment, therefore, obtains an independent encoder representation of the document, which when fused with the other modules, is fed to the BART decoder to generate the final summary.

## 3.2 Simplicial Complex Layer

Summarizing documents does not always entail information in pairwise connections. Consequently, methods based on single-dimensional graphs fall short in capturing the comprehensive context and its intricate relationships. Conversely, the utilization of a simplicial complex layer enables the incorporation of higher-order graphs, where the inclusion of simplices (vertices within a single-order graph) empowers the modeling of information that extends beyond pairwise associations.

A simplicial complex is a mathematical structure that represents relationships between different levels of "simplices." A simplex is a generalization of a triangle (2-simplex) to higher dimensions. In the context of summarization, simplicial complexes are used to capture relationships between multiple documents that might not be directly connected, by considering their shared properties or interactions. This approach formulates an exceedingly diverse and extensive feature set, facilitating the model's ability to comprehend inter-document relations in a more comprehensive and nuanced manner.

We use the simplicial complex (SC) layer in the self-attention setting similar to (Giusti et al., 2022).

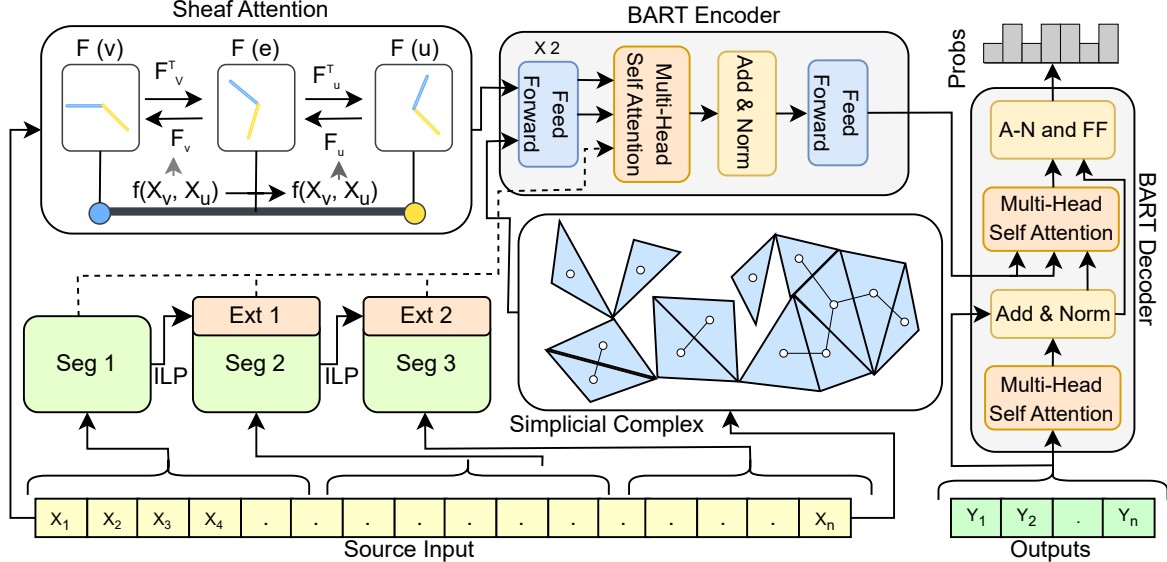

Figure 1: A schematic architecture of FABRIC. We adept the BART encoder and introduce simplicial complex layer and sheaf graph attention and fuse them with multi-head attention of BART. The fused representation is passed to the BART decoder to generate candidate summaries.

Mathematically, we define the input to SC layer as $SC_i$ and learnable weights as $W_{sc}$. The transformations are applied to the lower irrotational filter $H^i$, and upper solenoidal filter $H^s$. The transformations are represented by $H^i = [SC_i]_i W_{sc}^{(i)} \in R^{F_{l+1}}$, and $H^s = [SC_i]_i W_{sc}^{(i)} \in R^{F_{l+1}}$. The representation $H^i$ and $H^s$ are combined for each iteration, and self-attention is computed using the following equations,

$$e_{l,i,j}^{(u)} = a_l^{(i)}\left(h_{l,p}^{(i)}, h_{l,q}^{(i)}\right) \qquad \text{for } j \in \mathcal{N}_i^{(u)},$$
$$e_{l,i,j}^{(d)} = a_l^{(s)}\left(h_{l,p}^{(s)}, h_{l,q}^{(s)}\right) \qquad \text{for } j \in \mathcal{N}_i^{(d)}$$

These representations are normalized using the softmax function as $\alpha_l^{(u)} = \text{softmax}_j(e_i^{(u)})$, and $\alpha_l^{(s)} = \text{softmax}_j(e_l^{(s)})$. The attention weights of the SC layer are computed using,

$$Z_{l+1} = \sigma_l\left(\sum_{p=1}^{J_l^{(d)}} \left(L_l^{(d)}\right)^p Z_l W_{l,p}^{(d)} + \sum_{p=1}^{J_l^{(u)}} \left(L_l^{(u)}\right)^p Z_l W_{l,p}^{(u)} + \widehat{P}_l Z_l W^{(h,l)}\right)$$

where the coefficients of the upper and lower attentional Laplacians, $\mathbf{L}_l^{(u)}$ and $\mathbf{L}_l^{(d)}$, respectively are obtained . The filter weights $\left\{\mathbf{W}_{l,p}^{(d)}\right\}_p$, $\left\{\mathbf{W}_{l,p}^{(u)}\right\}_p$, $\mathbf{W}_l^{(h)}$ and the attention mechanism parameters $\mathbf{a}_l^{(u)}$ and $\mathbf{a}_l^{(d)}$ are learnable parameters, while the order $J_l^{(d)}$ and $J_l^{(u)}$ of the filters, the number $F_{l+1}$ of

output signals, and the non-linearity $\sigma_l(\cdot)$ are hyperparameters to be chosen at each layer. The final representation is passed through a feed-forward layer.

### 3.3 Sheaf Graph Attention

Similar to graph neural networks (GNNs) (Scarselli et al., 2009) and graph attention networks (GATs) (Veličković et al., 2018), the contextual representation of simplicial complex (SC) layer encounters limitations in performance, particularly in heterophilic settings and over-smoothing. Moreover, relying on a single module to capture the inter-document relationship may introduce biases in representing factual knowledge. To overcome this limitation, we introduce a new module, called sheaf graph attention in the MDS setting, taking inspiration from the prior sheaf attention networks (Hansen and Gebhart, 2020).

A sheaf is a mathematical construct used to associate local data with certain topological spaces. In a sheaf, data are organized in a way that respects the local structures of the underlying space. In the context of graphs, sheaf graphs extend the idea of traditional graph structures by incorporating not only node and edge connections but also the information associated with these connections. In practical terms, sheaf graphs can represent relationships between data points while also encoding additional context or attributes associated with those

relationships. This makes them suitable for capturing complex interactions, semantic relationships, and context in various applications, including natural language processing tasks like summarization.

Mathematically, we define the input vector as $X_s$ and introduce the learnable weight matrices $W_a$ and $W_b$. The identity matrix is denoted by $I$, and the sheaf Laplacian is represented as $\psi$. We perform a Kronecker product between $I$ and $W_a$. The sheaf Laplacian $\psi$ is defined as a vector in an undirected graph $G_u$, where vertices are defined by $v$ and edges by $e$. Mathematically, it can be expressed as follows:

$$Sheaf(X) = relu((I - \psi)(I \otimes W_a)XW_b) \quad (1)$$

Here, X is the data transformation block, and $\otimes$ is Kronecker product.

### 3.4 Encoder Setting

The first module introduces a document segments via topic guidance, which enables FABRIC to generate multiple target sequence vectors with segmented source documents aided with oracle proxies. The sequence vectors are learned using the BART encoder model. These feature vectors helps the model to understand the linguistic properties and retain multiple context vectors. The second module introduces the simplician complex layer, helping the model to understand the inter-document relations and aiding the weights of the vectors to promote the factualiy and faithfulness. As the feature vector from the SC layer are not same as attention input, the output is passed through a feed-forward layer and fused with the multi-head attention module of BART. Finally, the vectors from the sheaf graph attention model also undergo normalization via a feed-forward layer for feature vector normalization, followed by fusion with the heads of the multi-head attention in the BART encoder. The SC layer and sheaf graph contribute to the model's understanding of inter and intra-document relationships and the weighting of contexts, thereby promoting factual and faithful representations for the decoder generator model.

### 3.5 Decoder Setting

The amalgamated representation derived from the encoder module of FABRIC, consisting of the BART encoder, SC layer, and sheaf graph attention, is forwarded to the BART decoder module. In this configuration, the BART decoder generates summaries utilizing the encoder representation and the pointing mechanism. Much like a pointer generator, the pointing mechanism enables the model to directly copy text from the source document.

## 4 Datasets

We benchmark our study on four widely popular datasets. (i) **Multinews** (Fabbri et al., 2019) comprises news articles as source documents and human-written summaries as target summaries. (ii) **CQASumm** (Chowdhury and Chakraborty, 2019) dataset consists of question-answer pairs, where the accepted answer is used as the target summary and the remaining answers form the source document. (iii) **DUC** (DUC, 2002) includes news articles as source documents, with target summaries manually annotated by editors. (iv) **Opinosis** (Ganesan et al., 2010) combines user reviews from various platforms, where individual opinions serve as the source document, while the target summaries are human-annotated.

## 5 Abstractive Baselines

(i) The **Pointer Generator (PG)** approach (See et al., 2017) combines attention, and pointing mechanism to apprehend inter-document relationships. (ii) **Himap** (Fabbri et al., 2019) amalgamates MMR sentence weights in the PG network to emphasize pivotal information in the summary. (iii) **Bottom-up Transformers** (Gehrmann et al., 2018) employ the Transformer architecture (Vaswani et al., 2017) with one random attention head functioning as the copy pointer. (iv) **BERT** (Devlin et al., 2019) is an encoder-only language model trained using token masking techniques. (v) **BART** (Lewis et al., 2020) employs an encoder-decoder model trained on the text span masking technique. (vi) **T5** (Raffel et al., 2020), akin to BART, is an encoder-decoder-based model that treats all downstream tasks as text-to-text problems during training. (vii) **LongT5** (Guo et al., 2021) scales the T5 architecture and utilizes sentence masking during pretraining to enrich language generation. (viii) **Pegasus** (Zhang et al., 2020) adopts an innovative sentence masking technique in the language model to capture sentence-level representations. (ix) **Longformer** (Beltagy et al., 2020) incorporates a Transformer with sparse attention to handle elongated sequences.(x) **BRIO** (Liu et al., 2022) employs a stochastic approach, rather than maximum likelihood, to train the network.(xi) **PRIMERA** (Xiao et al., 2022) extends

| System | Multinews | | | CQASumm | | | DUC | | | Opinosis | | |
|---|---|---|---|---|---|---|---|---|---|---|---|---|
| | R-1 | R-2 | R-L | R-1 | R-2 | R-L | R-1 | R-2 | R-L | R-1 | R-2 | R-L |
| PG | 41.85 | 12.91 | 20.79 | 31.09 | 5.52 | 21.85 | 31.43 | 6.03 | 23.08 | 19.65 | 1.29 | 20.08 |
| HiMAP | 44.17 | 16.05 | 24.38 | 27.13 | 4.48 | 19.87 | 31.44 | 6.11 | 23.17 | 18.02 | 1.46 | 19.84 |
| Transfor. | 44.32 | 15.11 | 28.07 | 27.52 | 4.53 | 21.0 | 32.14 | 6.24 | 23.34 | 20.46 | 1.41 | 21.35 |
| BERT | 44.27 | 16.23 | 31.41 | 27.32 | 4.55 | 21.14 | 34.64 | 6.39 | 24.18 | 20.41 | 4.62 | 21.84 |
| BART | 48.47 | 18.41 | 33.21 | 27.84 | 4.65 | 21.74 | 35.41 | 6.48 | 24.67 | 19.8 | 6.82 | 26.87 |
| T5 | 44.08 | 16.39 | 37.68 | 28.11 | 3.74 | 21.45 | 34.13 | 6.32 | 24.13 | 27.41 | 6.97 | 27.41 |
| LongT5 | 48.17 | 19.43 | 38.94 | 28.74 | 4.84 | 22.12 | 34.58 | 6.37 | 24.32 | 29.46 | 7.04 | 28.64 |
| Pegasus | 41.79 | 16.58 | 39.72 | 28.24 | 4.87 | 22.41 | 34.61 | 6.41 | 24.71 | 31.28 | 7.15 | 29.39 |
| Longfor. | 46.89 | 18.50 | 41.83 | 28.01 | 5.05 | 23.86 | 36.31 | 6.76 | 27.11 | 33.32 | 8.63 | 30.49 |
| BRIO | 47.24 | 19.34 | 44.57 | 29.13 | 5.12 | 24.17 | 37.18 | 7.02 | 29.37 | 31.15 | 8.52 | 29.64 |
| PRIMERA | 49.90 | 21.11 | 46.23 | 31.54 | 5.58 | 26.57 | 37.84 | 7.21 | 31.18 | 34.64 | 9.12 | 33.20 |
| **FABRIC** | 50.68 | 21.26 | 46.57 | 35.81 | 6.91 | 31.47 | 39.64 | 8.75 | 34.12 | 39.21 | 11.42 | 35.48 |
| Δ - bsln | ↑0.78 | ↑0.11 | ↑0.34 | ↑4.27 | ↑1.33 | ↑4.90 | ↑1.80 | ↑1.54 | ↑2.94 | ↑4.57 | ↑2.30 | ↑2.28 |
| +SD & SC | 48.24 | 20.54 | 44.87 | 33.24 | 6.14 | 29.14 | 36.21 | 8.29 | 32.71 | 38.54 | 10.84 | 34.29 |
| +SC | 45.58 | 19.12 | 42.35 | 31.85 | 6.07 | 27.65 | 35.78 | 8.11 | 31.63 | 37.68 | 10.51 | 32.18 |
| +SD | 42.18 | 18.74 | 41.42 | 29.18 | 5.89 | 26.19 | 33.19 | 8.02 | 30.12 | 36.84 | 10.18 | 31.73 |

Table 1: Comparative analysis on four datasets – Multinews, CQASumm, DUC and Opinosis. We report ROUGE-1 (R-1), ROUGE-2 (R-2), and ROUGE-L (R-L) for eleven baselines. Our model FABRIC beats all the competing baselines.We also perform ablations over FABRIC. Addition of simplicial complex layer (SC) and Sheaf graph attention (SD) further improves the performance.

the Longformer architecture (Beltagy et al., 2020) by pretraining it on the downstream task.

# 6 Evaluation Setup

We benchmark FABRIC on the quantitative metrics – Rouge-1 (R1), Rouge-2 (R2) and Rouge-L (RL) (Lin, 2004), to evaluate the lexical overlap, as well as on qualitative metrics – BARTScore (Yuan et al., 2021) to compute the semantic overlap, and FactCC (Kryscinski et al., 2020), and SummaC (Laban et al., 2022) for faithfulness assessment.

## 6.1 Evaluation Metrics

To compare the quality of the summaries, we employ the following evaluation metrics. (i) **Rouge** computes the lexical overlap between the target and the generated summary. (ii) **BARTScore** computes the semantic overlap between target and the generated summary using the pre-trained BART (Lewis et al., 2020) LM. (iii) **SummaC** evaluates the consistency of summary sentences w.r.t the source article, taking into account any inconsistencies that may occur throughout the source text. (iv) **FactCC** leverages a model trained to assess the consistency between a text/summary and its source article using synthetic data generated through various transformations.

## 6.2 Human Evaluation Setup

We conducted a human evaluation to assess the qualitative aspects of the summaries generated by FABRIC. The evaluation focused on five parameters – Informativeness (Inf), Relevance (Rel), Coherence (Coh), Fluency (Flu), and Topic coherence (Topic). To ensure a representative evaluation, we randomly selected 50 test samples from the system-generated summaries and engaged 30 annotators[4] with expertise in the evaluation process. To minimize bias, each sample was annotated by at least two annotators, and annotations with a divergence of more than two degrees were excluded to maintain consistency. The assigned scores by the annotators were then averaged for each sample. For additional details, please refer to Appendix B.

# 7 Experimental Results

We perform quantitative and qualitative evaluation on the four datasets. We benchmark the datasets over eleven baselines and present quantitative and qualitative results.

## 7.1 Quantitative Evaluation

Table 1 presents the lexical overlap between the various generated summaries and the reference summaries. Our results demonstrate that FABRIC

---
[4]The annotators were linguistics/subject experts and their age ranged between 25-35.

| System | Multinews | | | CQASumm | | | DUC | | | Opinosis | | |
|---|---|---|---|---|---|---|---|---|---|---|---|---|
| | BS (↑) | FC (↑) | Sa (↑) | BS (↑) | FC (↑) | Sa (↑) | BS (↑) | FC (↑) | Sa (↑) | BS (↑) | FC (↑) | Sa (↑) |
| PG | -2.81 | 62.4 | 70.4 | -3.21 | 48.3 | 47.6 | -4.74 | 45.1 | 70.5 | -4.37 | 55.4 | 56.1 |
| HiMAP | -2.73 | 65.8 | 71.4 | -3.17 | 48.5 | 47.7 | -4.68 | 45.2 | 71.4 | -4.33 | 56.1 | 57.6 |
| Transf. | -2.43 | 67.8 | 73.8 | -3.11 | 49.8 | 48.1 | -4.21 | 47.1 | 72.1 | -4.02 | 58.4 | 59.6 |
| BERT | -2.66 | 73.1 | 79.8 | -3.13 | 51.7 | 49.7 | -4.37 | 49.6 | 74.2 | -4.28 | 60.1 | 62.8 |
| BART | -2.51 | 76.8 | 81.2 | -3.08 | 53.9 | 51.8 | -4.34 | 51.8 | 75.6 | -4.21 | 62.7 | 64.1 |
| T5 | -2.34 | 74.2 | 81.4 | -3.02 | 51.8 | 50.1 | -4.31 | 50.1 | 74.2 | -4.18 | 61.2 | 62.8 |
| LongT5 | -1.92 | 75.4 | 83.4 | -2.86 | 54.2 | 50.4 | -3.87 | 50.8 | 73.5 | -4.13 | 62.1 | 63.4 |
| Pegasus | -2.14 | 76.2 | 84.2 | -2.82 | 55.1 | 51.1 | -3.95 | 51.4 | 76.2 | -4.25 | 62.4 | 64.7 |
| Longf. | -1.89 | 76.8 | 86.7 | -2.73 | 56.2 | 52.9 | -3.71 | 52.9 | 77.8 | -4.02 | 62.7 | 65.8 |
| BRIO | -1.85 | 75.4 | 85.2 | -2.68 | 55.4 | 51.2 | -3.60 | 52.5 | 76.4 | -3.97 | 61.2 | 62.8 |
| PRIMERA | -1.73 | 77.3 | 87.1 | -2.62 | 57.1 | 53.8 | -3.58 | 53.6 | 78.7 | -3.86 | 63.4 | 66.5 |
| **FABRIC** | -1.71 | 78.1 | 88.2 | -2.60 | 57.6 | 54.2 | -3.57 | 54.9 | 79.1 | -3.77 | 64.8 | 68.3 |

Table 2: Performance of the competing method in terms of BARTScore (BS), FactCC (FC), and SummaC (Sa). Lower BS indicates better performance.

| System | Multinews | | | | | | CQASumm | | | | | |
|---|---|---|---|---|---|---|---|---|---|---|---|---|
| | Inf | Coh | Rel | Flu | Topic | BERTopic | Inf | Coh | Rel | Flu | Topic | BERTopic |
| BART | 3.13 | 2.84 | 2.17 | 2.37 | 2.31 | 2.31 | 2.28 | 1.92 | 2.35 | 2.14 | 2.52 | 1.47 |
| Longformer | 3.02 | 2.92 | 2.12 | 2.29 | 2.43 | 2.34 | 2.19 | 1.93 | 2.37 | 2.15 | 2.51 | 1.49 |
| PRIMERA | **3.27** | 3.12 | 2.94 | 2.48 | 2.47 | 2.37 | 2.24 | 2.33 | 1.97 | 2.17 | 2.53 | 1.50 |
| FABRIC | 3.25 | 3.16 | 2.94 | 2.49 | 2.49 | 2.39 | 2.32 | 2.35 | 2.02 | 2.17 | 2.59 | 1.53 |

Table 3: Scores for five human evaluation metrics - Informativeness (Inf), Relevance (Rel), Coherence (Coh), Fluency (Flu) and Topic Coherence (Topic) and one automatic metric - BERTopic over three baselines and our proposed model, FABRIC.

attains 50.68 R1 and 46.57 RL, beating the best baseline (PRIMERA) by +0.78 R1 and +0.34 RL points. For CQASumm, FABRIC attains 35.81 R1 and 31.47 RL beating PRIMERA by +4.27 R1 and +4.90 RL points. Similarly, for DUC and Opinosis, FABRIC attains 39.64 R1 and 34.12 RL, respectively and 34.12 R1 and 35.48 RL respectively, beating PRIMERA by +1.80 R1, +4.57 R1, +2.94 RL, and +2.28 RL respectively.

To demonstrate the effectiveness of each module in FABRIC, we conduct an ablation study. The results of each ablation are presented in Table 1. Our base model, augmented with the simplicial complex (SC) layer, enhances the R1 and RL scores compared to the multinews baseline by +2.89 R1 and +9.14 RL points, respectively. Furthermore, the integration of sheaf graph attention contributes to additional improvements, boosting the performance by +2.66 R1 points and +2.52 RL points. These findings corroborate our hypothesis that language models alone are insufficient to capture inter-document relationships. However, when combined with context captured on higher dimension, they significantly enhance the model's ability to achieve high performance.

## 7.2 Qualitative Evaluation

We also conducted a human evaluation to assess the performance of FABRIC. Table 3 presents the results, demonstrating that our model generates high-quality summaries that are redundant, faithful, and easily comprehensible to humans. In addition to the human evaluation, we employed the BARTScore (Yuan et al., 2021), FactCC (Kryscinski et al., 2020), and SummaC (Laban et al., 2022) metrics to evaluate the semantic overlap and faithfulness between the system-generated summaries and the reference summaries.

Table 2 provides an assessment of the semantic overlap, faithfulness, and factuality of the generated summaries. The results demonstrate a significant improvement in all three qualitative metrics. In terms of semantic overlap, FABRIC achieves a score of $-1.71$ compared to multinews, exhibiting an improvement of $+0.02$ over the best baseline, PRIMERA. Similarly, for the FactCC and SummaC metrics, FABRIC shows improvements of $+0.8$ and $+1.1$, respectively. For the CQASumm dataset, FABRIC achieves a semantic overlap of $-2.60$, a FactCC score of 57.6, and a SummaC score of 54.2, surpassing the best baseline, PRIMERA, by

| Model | Output |
|---|---|
| Target-1 | A teenage would-be bank robber in Detroit found himself imprisoned—and presumably regretting one or two life choices—within seconds of being handed money at a bank on Monday, police say. A police spokesman tells the Detroit Free Press that the 15-year-old gave a teller a note claiming he had a bomb and demanding money. [...] |
| FABRIC | A bank teller in Detroit managed to stop a would-be bank robber before he could make it to the sidewalk. Police say a 16-year-old boy walked into a Chase Bank on the city's east side Monday afternoon and told the teller he was strapped with a bomb and to give him all the money, reports the Detroit Free Press. The teller did, and the teen set off for the doorway, "likely exhilarated by the scent of escaping with his new wealth," reports.[...] |
| ChatGPT | Kelly Preston, in an interview with Natalie Morales on 'Today,' discusses the "silent birth" technique commonly practiced by Scientologists. This technique involves refraining from spoken words during childbirth to create a peaceful environment. Preston credits this technique for the calm and peaceful demeanor of her children. [...] |
| Target-2 | A preservationist in New York City thinks he's found the site of a 1750s tavern [...] Adam Woodward says he found what he believes to be hand-hewn and hand-planed joists and foundation walls in the basement. |
| FABRIC | So he searched the basement, and "found myself in what I am pretty certain is the 1750s historic tavern," he says. Specifically, he found what he thinks are hand-hewn and hand-planed joists and foundation walls from Colonial. |
| ChatGPT | Adam Woodward believes he has found evidence that a building in Manhattan could be the former site of the Bull's Head Tavern, a famous 18th-century tavern where George Washington is believed to have visited during the American Revolution. |

Table 4: Comparison of target summary with the summary generated by FABRIC and ChatGPT.

+0.02, +0.6, and +0.4 points, respectively. Table 2 summarizes the improvements across all baselines on four datasets. These findings confirm our hypothesis that language models alone are insufficient to comprehend the relationships between multiple entities, necessitating the incorporation of additional modules to ensure the accuracy and coherence of the generated summaries.

The quantitative improvements achieved by our model are further supported by human assessments. Table 3 presents the results of these assessments, revealing FABRIC's consistent performance across all datasets. With the exception of informativeness on the multinews dataset compared to the PRIMERA baseline, FABRIC achieves the highest scores in all metrics on multinews and CQAsumm. This indicates that the generated summaries are highly faithful, relevant, and coherent when compared to the other baselines. Although FABRIC exhibits some shortcomings in terms of informativeness according to the human evaluations, it still outperforms other baselines by a significant margin. A detailed examination of the generated summaries and an analysis of the findings can be found in Section 8.

## 8 Error Analysis

As indicated in Table 3, significant improvements are observed across all four metrics of human evaluation. The generated summaries successfully capture the essence of the source documents and cover the major topics discussed. For instance, in sample #1 in Table 4, the generated summary not only comprehensively conveys the main incident but also provides additional relevant information. This additional information is captured by the simplicial complex layer and the sheaf graph attention, which pass it on to the decoder for inclusion in the final summary. However, in the case of sample #2, the source document presents the information about the year 1750 as a hypothesis, while another document presents it as a quotation. This discrepancy leads FABRIC to treat this information as definitive and present it as a quoted fact in the summary. This highlights the challenge of properly handling such cases and emphasizes the notion that target summaries and quantitative metrics alone may not suffice as the true performance measure (Li et al., 2021) for the task of abstractive summarization.

## 9 FABRIC vs ChatGPT

We conducted a comparison between the summaries generated by FABRIC and ChatGPT as shown in Table 4. We randomly generated 50 summaries from the multinews test set for this analysis. When comparing the two, we found that the summaries generated by ChatGPT are often overly general, lacking relevance and informativeness in relation to the source documents. For instance, in sample #1, ChatGPT discusses a general topic that deviates from the main focus by expanding on its own knowledge graph. Although it showcases linguistic capabilities, it fails to align the generated summary with the specific factual information from the source document. Similarly, in sample #2, Chat-

GPT provides a summary that captures the main idea of the source but neglects to mention important factual details such as the year or any correlations with specific locations. This comparison highlights that general purpose LLMs like ChatGPT have a tendency to focus on linguistic aspects but struggle to ensure fidelity to the factual information and alignment with the original source. As a result, they can be considered unfaithful as they deviate from the source and expand the generated output based on their own knowledge.

## 10 Conclusion

In this study, we presented FABRIC, a encoder-decoder model designed to enhance topic coherence and inter-document relations for multi-document abstractive summarization. The key components of FABRIC include BART for capturing linguistic aspects and implicial complex Layer and sheaf graph attention for capturing inter-document relationships. We compared FABRIC with eleven baselines on four widely popular MDS datasets. The results consistently demonstrated that FABRIC surpasses existing systems and achieves significant improvements across both quantitative and qualitative measures. These findings were further backed by human evaluation, validating the effectiveness of FABRIC in generating more accurate and faithful summaries that better represent the content of the source document.

## 11 Acknowledgement

The authors acknowledge the support of ihub-Anubhuti-iiitd Foundation at IIIT-Delhi.

## 12 Limitations

While our study presents promising results for FABRIC in the context of multi-document abstractive summarization, there are certain limitations that should be acknowledged. Firstly, FABRIC demonstrated improvements in both quantitative and qualitative evaluation metrics, the subjective nature of summarization quality makes it challenging to establish a universally agreed-upon standard for evaluation. Secondly, the performance of FABRIC could be influenced by factors such as the size and quality of the training data, hyperparameter tuning, and architectural choices, which should be further investigated. Lastly, the computational complexity and resource requirements of FABRIC should be taken into consideration, as

they may limit its practical applicability in certain real-time or resource-constrained settings. Overall, while FABRIC shows promising advancements in multi-document abstractive summarization, further research is needed to address these limitations and extend its applicability to a broader range of scenarios.

## 13 Ethics Statement

The benchmarked datasets – Multinews, CQA-Summ, DUC and Opinosis are available under open-source licenses. The baselines were also covered by the open-source licenses. The human evaluation for the quality assessment of summaries were done by students from academic institute. All experiments were performed over Nvidia A100 (80GB) GPU.

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

# A  Appendix

## A.1  System Setup/Implementation Details

For Multinews and CQASumm , we used the original cleaned corpus and segmented the source doucments as per the top-5 topics coverage. We trained the BERTopic over the source corpus and generated topics later. On average, we segmented the source document into three documents as per the upper limit of BART encoder and number of topics found. For DUC and Opinosis, we used the model trained on the multinews dataset for inference. For modeling, we used PyTorch 1.14 for prototyping. We initially set the learning rate to 0.002, and later warm-up is applied for the first 10000 steps. The model is trained for 8 epochs for multinews and CQASumm.

# B  Human Evaluation

We evaluate the qualitative aspects of the generated summaries on 5 metrics – Informativeness, Relevance, Coherence, Fluency, and Topical modeling. We define these metrics as follows:

**(1) Informativeness:** The degree to which a summary provides accurate and comprehensive information about the source text, conveying the main points effectively. **(2) Relevance:** The extent to which a summary is directly related and applicable to the topic and content of the source text, capturing the key aspects and avoiding irrelevant details. **(3) Coherence:** The coherence of a summary refers to its logical flow and smooth organization of ideas, ensuring that the sentences and paragraphs are well-connected and cohesive. **(4) Fluency:** Fluency measures the readability and naturalness of a summary's language, including grammar, syntax, and smoothness of expression, allowing for easy comprehension and readability. **(5) Topical modeling:** The overarching subject or theme that the summary focuses on, capturing the main idea and central topic of the source text, while maintaining coherence and relevance in conveying the essential information.

We also show few more examples of our proposed FABRIC in Table 5.

## C Discussions

**Why not use traditional graphs:** To benchmark with the traditional graph-based methods, we utilized (Yasunaga et al., 2017) to model the interconnections between the documents. However, when tested, the performance improvement was only seen for the DUC dataset. For Multinews, CQA-Summ and Opinosis, the performance was inconsistent over the syntactical and semantic metrics. We infer that traditional graph-based approaches struggle to capture nuanced dependencies and higher-order connections between the documents. In contrast, Simplicial Complex and Sheaf Graph capture the underlying semantics and meaning of the documents, not just their surface-level links. It provides a way to incorporate semantic context and relevance into the model, facilitating a more holistic understanding of how information is distributed across the documents.

**How do Sheaf Graphs improve performance in heterophilic settings and counter over-smoothing:** In a heterophilic setting, where nodes have varying degrees of relevance and connections across different subgroups or categories, traditional graph attention mechanisms might struggle to capture diverse relationships effectively. Sheaf graph attention excels in this scenario by allowing for adaptive information flow based on the local structures. Unlike standard graph attention, which computes weights based on node similarities, sheaf graph attention can tailor its attention weights for each context, considering both local and global patterns. This ability to differentiate and adapt to heterophilic relationships enables better representation of information spread across different subgroups, leading to improved performance. Over-smoothing is a common issue in graph neural networks (GNNs), where nodes' representations become overly similar after multiple graph convolutions, causing loss of discriminative information. Sheaf graph attention mitigates over-smoothing by incorporating higher-order relationships and context-specific attention. Instead of relying solely on aggregating neighbouring nodes' features, sheaf graph attention considers simplicial structures and focuses on more complex interactions, preventing overemphasis on similar nodes. This allows nodes to maintain their distinct characteristics even after multiple iterations of information propagation.

| Model | Output |
|---|---|
| Target | The unemployment rate ticked down from 5.9% to 5.8% last month, its lowest level since July 2008, the AP reports. And 214,000 jobs were added; 233,000 had been expected, per the Wall Street Journals forecast, with the unemployment rate expected to stick at 5.9%. August and September numbers were also revised upward: [...] |
| FABRIC | The economy added 214,000 jobs in October, below the 233,000 economists expected, but the unemployment rate dropped to 5.8% from 5.9%, reports the Wall Street Journal. That's the lowest rate since July 2008, notes the AP. The Labor Department also added 31,000 more jobs in August and September than previously estimated. Employers have now added at least 200,000 Jobs for nine straight months, the longest such stretch since 1995. [...] |
| Target | The US won't be competing in men's soccer at the Olympics, thanks to a heartbreaking goal that came literally at the last minute. El Salvador battled the US to a 3-3 draw last night, the AP reports; the US, which had been heavily favored to win its group in CONCACAF qualifying, needed a win after being upset by Canada 2-0. It was a back-and-forth affair, with the US scoring in the first minute, then falling behind, and finally coming back to lead 3-2. |
| FABRIC | The US mens soccer team is out of Olympic qualifying for the 2012 London Games after a 3-3 tie with El Salvador last night. The Americans led 3-2 in the final minute of stoppage time, but substitute keeper Sean Johnson couldnt handle a long shot from El Salvadors Jaime Alas, and the ball bounced off his hands, up over him, and into the net, reports AP. The tie means the US will miss the Olympics for the second time in three games. [...] |
| Target | If this story didnt already cause you to develop a mild case of gephyrophobia, then this one may do the trick: Some 11% of Americas bridges are structurally deficient and in need of repair, according to a new report from Transportation for America. Its a stat made more serious when you consider this next one: 260 million trips are made across the 66,405 problematic bridges each day. |
| FABRIC | More than one in nine bridges in the US is structurally deficient, meaning its in danger of collapse, according to a new report from Transportation for America, and thats a problem thats only going to get worse as the nations most heavily traveled bridges near the end of their expected life spans, USA Today reports. "You're seeing the aging of the system," says a co-author of the report. "It really does parallel the (aging of) the Baby Boomers in a startling way." The report found that at least 66,405 bridges—11% of the total—are at least 65 years old on average, [...] |

Table 5: Comparison of target summary with our FABRIC model generated summaries.