# OpenReview forum: "Promoting Topic Coherence and Inter-Document Consorts in Multi-Document Summarization via Simplicial Complex and Sheaf Graph"
_EMNLP/2023/Conference — EMNLP 2023 Main_

### Official Review · Reviewer_EADd · 2023-08-04

**Soundness:** 4

**Excitement:**

3: Ambivalent: It has merits (e.g., it reports state-of-the-art results, the idea is nice), but there are key weaknesses (e.g., it describes incremental work), and it can significantly benefit from another round of revision. However, I won't object to accepting it if my co-reviewers champion it.

**Paper Topic And Main Contributions:**

The paper proposes a new model for multi-document summarization using (1) simplicial complex layers and (2) sheaf graph attention. Along with this model, the paper suggests obtaining contextual vectors for any input length.

The authors show that the proposed model consistently outperforms 11 models on 4 popular MDS datasets according to automatic metrics (lexical and model-based metrics). In addition, the authors conduct a human evaluation, showing similar trends.

**Reasons To Accept:**

A new model for multi-document summarization that consistently outperforms state-of-the-art models on 4 benchmarks according to automatic metrics.

**Reasons To Reject:**

The paper claims that FABRIC improves faithfulness and factuality. However, this claim is not assessed in the results because the automatic faithfulness metrics (i.e., Summac and FactCC) were used between the generated summary and the reference, and the human evaluation also doesn’t assess faithfulness.

Regarding the human evaluation:


- What was the IAA for the human evaluation task? Also, the authors mentioned in lines 439-441 that “annotations with a divergence of more than two degrees were excluded to maintain consistency”. A large divergence between annotators might also reflect that the instructions are subjective or not sufficiently clear or that the annotators didn’t understand the task. How many cases were excluded because of a large divergence?


- From the definition of informativeness: “The degree to which a summary provides accurate and comprehensive information about the source text”, relevance: “The extent to which a summary is directly related and applicable to the topic and content of the source text” and topic modeling: “The overarching subject or theme that the summary focuses on, capturing the main idea and central topic of the source text”, one can understand that annotators were given all the source documents and asses the generated summary to the entire set of input documents. Is this the case or the generated summary was only compared to the reference? If so, the definitions should be updated accordingly.


- Overall, the improvement is rather minor compared to PRIMERA (0.02 on average for all evaluation aspects).


The comparison to ChatGPT in Section 9 is a bit misleading, showing some cherry picking examples in Table 4. Who conducted the comparison? Did the annotators compare the generated summary to the source articles or to the reference? What was the IAA between the annotators? What are the criteria of evaluation? Are they the same as the human evaluation in Section 6.2?

**Reproducibility:**

3: Could reproduce the results with some difficulty. The settings of parameters are underspecified or subjectively determined; the training/evaluation data are not widely available.

**Reviewer Confidence:**

3: Pretty sure, but there's a chance I missed something. Although I have a good feel for this area in general, I did not carefully check the paper's details, e.g., the math, experimental design, or novelty.

**Typos Grammar Style And Presentation Improvements:**

Line 130: input length input → input length
Line 146: primera
Line 246: segment generated —> generated segment
Line 248: “LM to process” ?
Line 359: Datasets
Line 364: Missing references to CQASumm
Line 480: do you mean table 3?
Line 940: , instead of .


Sometimes, the authors write primer, sometimes primera, the actual name is primera.

Some BG about simplicial and sheaf graphs is needed

The third contribution is more a requirement than a contribution. For me, the paper proposes two contributions, (1) a method for processing multiple documents by compressing documents and (2) a new encoder-decoder model for MDS based on simplicial complex and sheaf graphs.

---

> ### Author Rebuttal · Authors · 2023-08-28
>
> **Human evaluation IAA and divergence assessment:** We computed Cohen's Kappa score to determine the Inter-Annotator Agreement (IAA), resulting in an aggregated Kappa score of 0.8731. Regarding divergence, 7.82% of annotators exhibited discrepancies. To address this, we reassigned these summary samples to new human evaluators, considering agreements exceeding 75%. At the final evaluation, 3.49% of samples were excluded to ensure consistency in human evaluation.
>
> **Definition of Informativeness, Relevance, and Topic Modelling:** You are correct. The source documents and the generated summaries were provided for the human evaluations. We will mention this explicitly in the revised paper.
>
> **Improvement in human evaluations across datasets:** In terms of automatic evaluations, when compared to PRIMERA, Multinews demonstrates an improvement of +0.78 R1 and +0.34 RL. CQASumm shows gains of +4.27 R1 and +4.90 RL. Similarly, for DUC and Opinosis, improvements of +1.80 R1, +2.94 RL and +4.57 R1, +2.28 RL are achieved, respectively. Regarding human evaluations, the benchmarks were conducted on random samples, revealing consistent improvements across metrics and datasets. This suggests that the proposed model exhibits strong generalization and consistent performance across diverse datasets.
>
> **ChatGPT vs. proposed model evaluations:** Due to space constraints, we presented limited examples; however, we plan to provide all samples upon the final paper's release. Regarding the comparison, human evaluators were provided with the source document, ChatGPT-generated summary, and proposed model-generated summary. The evaluation yielded an aggregated kappa score of 0.8382. The evaluation criteria were akin to those outlined in section 6.2.
>
> **Typos and Grammars:** We apologise for the typos and grammatical mistakes. We have gone through the paper and have revised it. We have also corrected the baseline name — PRIMERA.
>
> **Background about Simplicial Complex and Sheaf Graphs:** Thank you for the suggestion. We will include the motivation and background behind using Simplicial Complex and Sheaf Graphs in the main text. We will also append the supplementary with the required mathematical background.

---

### Official Review · Reviewer_W8oR · 2023-08-04

**Soundness:** 4

**Excitement:**

4: Strong: This paper deepens the understanding of some phenomenon or lowers the barriers to an existing research direction.

**Paper Topic And Main Contributions:**

This paper proposes FABRIC, a new model for multi-document summarization. It utilizes a simplicial complex layer to understand complex relationships, and sheaf graph attention to capture diverse properties. FABRIC outperforms other baselines.

**Questions For The Authors:**

1. Why do you use simplicial complex and sheaf graph to model the innerconnections of the given documents? Why not use traditional graphs or hypergraphs?
2. Why does the proposed sheaf graph attention improve the performance in heterophilic setting and also alleviate the over-smoothing problem?
3. In these MDS dataset used by this paper, the average number of input documents per sample is low. Authors may consider using datasets [1,2] with a higher average number of documents per sample to verify the effectiveness (Topic Coherence and Inter-Document Consorts) of the proposed method.

[1] Multi-XScience: A large-scale dataset for extreme multi-document summarization of scientific articles
[2] Generating a Structured Summary of Numerous Academic Papers: Dataset and Method

**Reasons To Accept:**

1. The paper adopts the simplical complexes structure to model the interconnections of multiple documents and also the sheaf graphs also facilitate the heterogeneous setting.
2. The experiment shows superiority in multiple document summarization task compared with other 11 baselines.
3. The paper is well-organized.

**Reasons To Reject:**

1. The equations are not well explained. For instance, in line 274, it did not explain the meaning of F. Please check the rest.
2. The graph construction method is not clear.
3. Ablation study should be conducted to reveal the contribution of each module.
4. The writing needs improvements. More method and rationale details are suggested to be given.

**Reproducibility:**

5: Could easily reproduce the results.

**Reviewer Confidence:**

5: Positive that my evaluation is correct. I read the paper very carefully and I am very familiar with related work.

---

> ### Author Rebuttal · Authors · 2023-08-28
>
> **Missing notations in equations:** Thank you for your suggestion. In line 274, the symbol "F" corresponds to a bank of Simplicial Filters. Furthermore, we have validated and updated all the notations used in the algorithm. We will update these in the updated version of the paper.
>
> **Graph construction method is unclear:** We do not explicitly construct a graph; instead, we pass the token embeddings ‘Xs' into a Sheaf Graph function to formulate a graph representation of the features, which are learned by the module during the training phase. This is similar to graph formulation and learning in the GCN and GAT approaches.
>
> **Ablations of proposed modules:** We performed ablation analyses for every module, as shown in Table 1 (main paper). The proposed Sheaf Graph (SD) module achieves a Rouge-1 score of 29.18 on CQASumm, 33.19 on DUC, and 36.84 on Opinosis. Further, the introduction of Simplicial Complex enhances the performance of Rouge-1 to 31.85 on CQASumm, 35.78 on DUC, and 37.68 on Opinosis.
>
> **Motivation and rationale behind using Simplicial Complex and Sheaf Graphs:** Thank you for this suggestion. We will include the following  motivation and background on Simplicial Complex and Sheaf Graphs in the main paper and append supplementary sections with the required mathematical foundation needed.
>
> `Simplicial Complex: A simplicial complex is a mathematical structure that represents relationships between different levels of "simplices." A simplex is a generalization of a triangle (2-simplex) to higher dimensions. In the context of summarization, simplicial complexes are used to capture relationships between multiple documents that might not be directly connected, by considering their shared properties or interactions.`
>
> `Sheaf graphs: A sheaf is a mathematical construct used to associate local data with certain topological spaces. In a sheaf, data are organized in a way that respects the local structures of the underlying space. In the context of graphs, sheaf graphs extend the idea of traditional graph structures by incorporating not only node and edge connections but also the information associated with these connections. In practical terms, sheaf graphs can represent relationships between data points while also encoding additional context or attributes associated with those relationships. This makes them suitable for capturing complex interactions, semantic relationships, and context in various applications, including natural language processing tasks like summarization.`
>
>
> **Why not use traditional graphs and hypergraphs:** To benchmark with the traditional graph-based methods, we utilized [1] to model the interconnections between the documents. However, when tested, the performance improvement was only seen for the DUC dataset. For Multinews, CQASumm and Opinosis, the performance was inconsistent over the syntactical and semantic metrics. We infer that traditional graph-based approaches struggle to capture nuanced dependencies and higher-order connections between the documents. In contrast, Simplicial Complex and Sheaf Graph capture the underlying semantics and meaning of the documents, not just their surface-level links. It provides a way to incorporate semantic context and relevance into the model, facilitating a more holistic understanding of how information is distributed across the documents. As for the hypergraphs, we did not run any ablations but theorized that hyperedges, when connected to multiple nodes, can lead to difficulties in interpreting the semantic relationships they represent. In abstractive summarization, understanding the precise meaning and context of these relationships is crucial. Secondly, Hypergraphs can become increasingly complex as the number of nodes and hyperedges grows.
>
> **How do Sheaf Graphs improve performance in heterophilic settings and counter over-smoothing:** In a heterophilic setting, where nodes have varying degrees of relevance and connections across different subgroups or categories, traditional graph attention mechanisms might struggle to capture diverse relationships effectively. Sheaf graph attention excels in this scenario by allowing for adaptive information flow based on the local structures. Unlike standard graph attention, which computes weights based on node similarities, sheaf graph attention can tailor its attention weights for each context, considering both local and global patterns. This ability to differentiate and adapt to heterophilic relationships enables better representation of information spread across different subgroups, leading to improved performance. Over-smoothing is a common issue in graph neural networks (GNNs), where nodes' representations become overly similar after multiple graph convolutions, causing loss of discriminative information. Sheaf graph attention mitigates over-smoothing by incorporating higher-order relationships and context-specific attention. Instead of relying solely on aggregating neighbouring nodes' features, sheaf graph attention considers simplicial structures and focuses on more complex interactions, preventing overemphasis on similar nodes. This allows nodes to maintain their distinct characteristics even after multiple iterations of information propagation.
>
> **Benchmarking with additional datasets:** Thank you for your suggestion. We conducted benchmarking on these datasets using our proposed model, and the results are summarized in the table below. Unfortunately, due to time constraints, we were unable to run the baselines. However, we will include the baseline results for these datasets in the revised paper.
>
> | Dataset        | Rouge-1    | Rouge-2    | Rouge-L    |
> |----------------|-------|-------|-------|
> | Multi-XScience | 35.82 | 7.81  | 31.97 |
> | BigSurvey-MDS  | 45.13 | 13.62 | 18.39 |
>
> [1] Michihiro Yasunaga, Rui Zhang, Kshitijh Meelu, Ayush Pareek, Krishnan Srinivasan, and Dragomir Radev. 2017. Graph-based Neural Multi-Document Summarization. In Proceedings of the 21st Conference on Computational Natural Language Learning (CoNLL 2017), pages 452–462, Vancouver, Canada. Association for Computational Linguistics.

---

### Official Review · Reviewer_PARH · 2023-08-11

**Soundness:** 4

**Excitement:**

2: Mediocre: This paper makes marginal contributions (vs non-contemporaneous work), so I would rather not see it in the conference.

**Paper Topic And Main Contributions:**

This work introduces FABRIC, a multi-document summarizer.  FABRIC consist of a BART encoder decoder model, augmented with a simplicial complex and topic segmentation components. The proposed method is comprehensive and well evaluated. Authors report both quantitative and qualitative evaluations.

**Reasons To Accept:**

- The paper is generally well structured, motivated
- Good experimental setup and ample evaluation
- Improvement over state of the art.

**Reasons To Reject:**

- It would have been good to evaluate the topic segmentation on its own.
- The proposed method is very difficult to understand for intended audience for this work (EMNLP attendees). One could imagine many NLP practitioners to be confused about the proposed method. I myself am unable to properly critique the work. Given the nature of the presented method, perhaps a venue such as NeurIPS would be more appropriate

**Reproducibility:**

4: Could mostly reproduce the results, but there may be some variation because of sample variance or minor variations in their interpretation of the protocol or method.

**Reviewer Confidence:**

1: Not my area, or paper was hard for me to understand. My evaluation is just an educated guess.

---

> ### Author Rebuttal · Authors · 2023-08-28
>
> **Evaluating topic segmentation:** Thank you for the suggestion. We employ topic segmentation to divide the document into multiple segments during preprocessing. In relation to this module, we do not evaluate this module due to the absence of gold labels. Instead, we assess the performance of the topic segmentation within the context of summarization, utilizing an automated metric (BERTopic), which is detailed in Table 3.
>
> **Is EMNLP the right venue?** The task of summarization closely aligns with the area of EMNLP, and we believe that EMNLP is the right platform for presenting this research. We will simplify the methodology section and add further background and motivation in the revision.

---

### Meta-Review · Area_Chair_xGec · 2023-09-13

**Recommendation:** 4

**Metareview:**

The paper tackles multi-document summarisation, by introducing a novel idea that uses simplicial complex and sheaf graph attention to capture complex and diverse cross-document relationships. Reviewers all thought the paper is generally well-structured, the experiments are extensive, and the results are impressive. Some concerns are raised by the reviewers, and the two main ones that I can see are: (1) certain parts surrounding the methodology are difficult to follow, and they require more description (e.g. explaining the idea of simplicial complex) and motivation (e.g. why do we need simplicial complex?); and (2) the human evaluation results require some clarification - in particular its inter-annotator agreement and the conclusion drawn from the ChatGPT results. That said, I believe the contribution of the paper is significant, and that these concerns are presentational issues that could be fixed in the revision.

---

### Decision · Program_Chairs · 2023-10-07

**Decision:**

Accept-Main

**Comment:**

The paper tackles multi-document summarisation, by introducing a novel idea that uses simplicial complex and sheaf graph attention to capture complex and diverse cross-document relationships. Reviewers all thought the paper is generally well-structured, the experiments are extensive, and the results are impressive. Some concerns are raised by the reviewers, and the two main ones that I can see are: (1) certain parts surrounding the methodology are difficult to follow, and they require more description (e.g. explaining the idea of simplicial complex) and motivation (e.g. why do we need simplicial complex?); and (2) the human evaluation results require some clarification - in particular its inter-annotator agreement and the conclusion drawn from the ChatGPT results. That said, I believe the contribution of the paper is significant, and that these concerns are presentational issues that could be fixed in the revision.